

# Expanding the coverage of MISR aerosol retrievals over shallow, turbid, and eutrophic waters

Robert R. Nelson[1], Marcin L. Witek[1], Michael J. Garay[1], Michael A. Bull[1], James A. Limbacher[2,3], Ralph A. Kahn[4], and David J. Diner[1]

[1]Jet Propulsion Laboratory, California Institute of Technology, Pasadena, CA, USA
[2]I. M. Systems Group, Inc. (IMSG), Rockville, MD, USA
[3]National Oceanic and Atmospheric Administration, College Park, MD, USA
[4]Earth Sciences Division, NASA Goddard Space Flight Center, Greenbelt, MD, USA

**Correspondence:** Robert R. Nelson (Robert.R.Nelson@jpl.nasa.gov)

**Abstract.**

Shallow and coastal waters are often rich in nutrients (eutrophic) and biologically productive, turbid from runoff, and located where the atmosphere above can be more aerosol-laden than over open ocean waters due to proximity to aerosol sources on land. Although the NASA Earth Observing System's Multi-angle Imaging SpectroRadiometer (MISR) on board the Terra

satellite has been monitoring global aerosols for over 23 years, the current operational retrieval algorithm (V23) is not applied over waters less than 50 m in depth or within 5 km of land, designated as "shallow water." This is due to the simplicity of the Dark Water algorithm, applied operationally over deep waters, which assumes the surface is essentially black in the primarily-used red and near-infrared spectral bands. In this work, we describe the implementation and validation of a "Shallow Water" aerosol retrieval algorithm for MISR, which takes advantage of all four available spectral bands and includes a Lambertian

surface reflectivity term to account for water-leaving radiance. This algorithm compares well to independent, surface-based observations and demonstrates better performance over shallow waters than the operational Dark Water retrieval algorithm. Globally, aerosol retrievals over shallow waters increase the total number of MISR over-water measurements by more than 7%, including new retrievals made over some of the most biologically productive parts of the ocean.

© 2023 California Institute of Technology. Government sponsorship acknowledged.

## 1 Introduction

The Multi-angle Imaging SpectroRadiometer (MISR) was launched onboard the NASA Earth Observing System (EOS) Terra satellite into a sun-synchronous, polar orbit on 18 December 1999. MISR was designed to monitor key parameters, such as aerosol amount and type, that describe the state of the Earth system (Kaufman et al., 1998). The instrument makes observations in four spectral bands (blue at 446.6 nm, green at 557.5 nm, red at 671.7 nm, near-infrared at 866.4 nm) using nine push-broom

cameras viewing at 0°, 26.1°, 45.6°, 60.0°, and 70.5° both fore and aft relative to the direction of satellite motion. A single point on the ground is thus viewed from nine different angles at four wavelengths over the span of approximately seven minutes. MISR has a common, cross-track swath width of ~380 km, providing nearly complete global coverage every nine days, with



more frequent observations near the poles. The instrument reports radiance measurements at spatial resolutions ranging from 275 m to 1.1 km, depending on the exact combination of camera and band (Diner et al., 1998). Although originally planned as a six-year mission, the instrument continues to make high-quality observations of Earth.


The initial operational MISR aerosol retrieval approach is described by Martonchik et al. (1998, 2002, 2009). Two different retrieval pathways are taken for locations determined to be over land or water. The focus of this work is the Dark Water (DW) retrieval, which is implemented over water surfaces. A description of this retrieval can be found in Kalashnikova et al. (2013) and Witek et al. (2018). The current operational version of the MISR aerosol product is Version 23 (V23), which is described in detail in Garay et al. (2020), with validation of the DW algorithm being discussed in Witek et al. (2019). The V23 Level 2 (L2) swath product reports aerosol information on a 4.4 km grid, including retrieval-level uncertainties and improved retrieval screening relative to previous versions.


A simplification in the operational DW algorithm is that it assumes the water surface to be nearly black in the red and near-infrared (NIR) bands primarily used in the retrieval. The small amount of water-leaving radiance emitted from deep, relatively clean waters (e.g., Morel and Maritorena, 2001) is accounted for with an underlight model (Kahn et al., 2005; Limbacher and Kahn, 2014, 2017). This underlight model is currently implemented as a fixed Lambertian surface reflection term with surface albedos of 2.57% (blue), 0.668% (green), 0.0930% (red), and 0.00635% (NIR) (Garay et al., 2020). However, there are many places on Earth where the water-leaving radiance – the upwelling radiant energy emerging from the water due to scattering within the water body and transmitted through the water surface quantified just above the surface (Mobley, 1999) – cannot adequately be accounted for by a simple underlight model (e.g., Sayer et al., 2010). Such locations include shallow, turbid, and eutrophic waters. In shallow waters, incident sunlight can reflect off the underwater surface. Turbidity occurs when particles, typically sediments from runoff, are suspended in the water and reflect light. Eutrophic waters are enriched with nutrients and can contain significant plant and algae growth that may reflect light. Moreover, water-leaving radiance itself is a fundamental parameter for ocean color remote sensing and provides information on the optical properties of the water contents, including chlorophyll-$\alpha$ concentration (Gordon et al., 1988; Mobley, 1999). To limit potential biases due to observations of non-black waters, the V23 DW retrieval is not performed over waters less than 50 m in depth or within 5 km of land. It is assumed that these cutoffs are sufficient to remove the majority of turbid and eutrophic waters, which preferentially occur near the coasts. These shallow coastal waters are some of the most biologically productive on Earth (Sigman and Hain, 2012). However, their representation in the global carbon budget still contains significant uncertainty (Bauer et al., 2013; Behrenfeld et al., 2005; Friedlingstein et al., 2020). Being able to better quantify aerosols over these coastal regions would improve atmospheric correction algorithms needed for satellite-based ocean color retrievals that could help reduce that uncertainty. These areas also typically have greater aerosol loadings than open ocean due to their proximity to aerosol sources over land.





In this work, we follow the approach introduced by Limbacher and Kahn (2019) and modify the operational DW retrieval algorithm to use all four MISR spectral bands in all situations and solve for a spectral surface Lambertian reflectance term meant to account for water-leaving radiance contributions. Other studies have also shown the potential of using MISR multi-angle observations to retrieve both the aerosol optical depth (AOD) and surface bidirectional reflectance. For example, a somewhat similar approach was employed by Han et al. (2022). However, their methodology differs significantly from the operational




MISR DW retrieval algorithm and their validation was limited in scope. Here, the modified retrieval algorithm is referred to as the "Shallow Water" (SW) retrieval algorithm. It differs from the Research Algorithm presented in Limbacher and Kahn
(2014, 2015, 2017, 2019), which is produced at a higher spatial resolution, contains advancements in radiometric calibration, and has an expanded aerosol optical model climatology. The Research Algorithm has only been implemented on a case-by-case basis, whereas the SW retrieval algorithm described here is implemented so as to be suitable to be included as part of the operational MISR aerosol retrieval processing chain.

This paper focuses on aerosol retrievals from MISR over shallow, turbid, and eutrophic waters and validating retrieved AOD
with collocated Aerosol Robotic Network – Ocean Color (AERONET-OC) surface-based measurements. Comparisons are made with both the DW and SW retrieval algorithms in order to demonstrate the improved performance of the SW algorithm relative to the current operational DW retrieval algorithm. Section 2 describes the data used in this work and the validation collocation criteria. Section 3 describes both the DW and SW retrieval algorithms, and Section 4 contains the AOD validation results, quality filtering development, three case studies, and remote-sensing reflectance validation for one of the case studies.
Finally, Section 5 discusses the results and presents the impact of using the SW retrieval algorithm globally to retrieve aerosols from MISR over shallow waters.

## 2  Data

The primary data used in this study were MISR L2 swath-based retrieval products, specifically AOD, reported at 550 nm on a 4.4 km by 4.4 km spatial grid ~380 km in width. The MISR L2 product requires several input parameters including the
measured radiances, calibration information, geographic information, and a land/water mask. Of importance for this work, the Ancillary Geographic Product (AGP) contains the MISR Digital Elevation Model (DEM), which provides the high-resolution surface elevations and land/water mask used in this work (Logan, 1999).

These MISR observations were collocated with data from the AErosol RObotic NETwork – Ocean Color (AERONET-OC, Zibordi et al., 2009, 2021). This subset of AERONET (Holben et al., 1998) provides both atmospheric aerosol information
and various metrics quantifying water-leaving radiance at multiple wavelengths (nominally 412, 443, 488, 531, 551, and 667 nm), which are useful for determining how well different MISR algorithms are able to retrieve AODs over scenes with significant water-leaving radiance. The AERONET-OC measurements are made by CE-318 and CE-318T sun-photometers installed on offshore platforms (e.g., lighthouses, service structures, and oil towers, Zibordi et al., 2021). The datasets provided by AERONET-OC used in this study were AOD and normalized water-leaving radiance ($L_{WN}$) corrected for bidirectional effects
($L_{WN}$ f/Q, where $f$ is a dimensionless factor that regulates the magnitude of the irradiance reflectance and $Q$ is a bidirectional function. See Morel et al., 2002).

The method used to compare to AERONET-OC followed the approach of Witek et al. (2019). Sites were collocated in time (within ±30 minutes) and space (within a 25 km radius circle). AERONET-OC observations were averaged if multiple collocations were available. Similarly, all the MISR retrievals within the collocation radius were averaged (up to ~100 retrievals).
Ultimately, 31 unique AERONET-OC sites were used in this work, with a total of 1579 collocations found in 1579 separate



MISR orbits from 23 April 2002 to 30 November 2020. The MISR data were then filtered on the cloud screening parameter (CSP), which is a measure of how many 1.1 km subregions and camera views within the retrieval region are designated by the retrieval algorithm as clear and likely to be suitable for an aerosol retrieval (Witek et al., 2013; Garay et al., 2020). The data were also filtered on CSP_3x3, which is the same as CSP but for a given pixel as well as the eight surrounding pixels. This is an additional metric designed to remove any residual cloud contamination. Of the 1579 initial AERONET-OC/MISR collocations identified, 1209 had at least one likely cloud-free MISR aerosol retrieval using the same cloud filtering as applied for the MISR operational aerosol product (CSP > 0.7, CSP_3x3 > 0.5, Garay et al., 2020).

In addition to the AERONET-OC validation dataset, 1339 MISR orbits spanning June through August 2016 were processed using the SW retrieval algorithm in order to determine the change in the number of retrievals when including measurements made over locations designated as shallow water (see Sect. 5 and Fig. 9).

## 3   MISR Retrieval Algorithm

The MISR aerosol retrieval algorithm starts with a look-up table (LUT) of pre-computed top-of-atmosphere (TOA) radiances. The current operational retrieval algorithm (V23) effectively contains 74 unique aerosol mixtures in the LUT (Garay et al., 2020). Each mixture can be made up of up to three unique particle types with prescribed optical and microphysical properties (Kahn et al., 2010; Kahn and Gaitley, 2015) from a total of eight primary types, including sea spray, sulfate, nitrate, mineral dust, carbonaceous, and urban soot aerosol optical analogs. The 74 mixture are intended to represent the majority of atmospheric aerosol scenes found on Earth. These pre-computed radiances are stored in the Simulated MISR Ancillary Radiative Transfer (SMART) LUT as equivalent reflectances:

$$\rho = \frac{\pi L}{E_0} \cdot D^2, \tag{1}$$

where $L$ is the TOA radiance, $D$ is the Earth-Sun distance at the time of observation in astronomical units (AU), and $E_0$ is the exo-atmospheric solar irradiance at 1 AU for each MISR spectral band. These SMART calculations include a modified linear mixing approach for mixtures containing more than one aerosol type (Abdou et al., 1997), an ozone correction, a Rayleigh scattering contribution, a polarization correction, and a sunglint and whitecap model driven by wind speed (Martonchik et al., 1998; Garay et al., 2020). SMART calculations are performed for each of the 74 mixtures and for 130 AOD values ranging from 0 to 3.

The modeled equivalent reflectances from SMART are then compared to the actual MISR measurements, which have been corrected for out-of-band and veiling light effects (Witek et al., 2017). The differences are minimized by calculating a cost function ($\chi^2_{\mathrm{abs}}$) to determine which set of modeled equivalent reflectances, and their corresponding aerosol amounts and properties, best match the actual measurements. For the current operational DW retrieval algorithm, this is done by calculating a cost function for each of the 74 aerosol mixtures and 130 AOD values as follows:





$$\chi^2_{\mathrm{abs,DW}}(\tau) = \frac{\sum\limits_{l=1}^{4} w_{\mathrm{DW}}(l) \cdot \left\{ \sum\limits_{j=1}^{9} v_{\mathrm{DW}}(l,j) \cdot \frac{[\rho_{\mathrm{MISR}}(l,j) - \rho_{\mathrm{m}}(l,j)]^2}{\sigma^2_{\mathrm{abs,DW}}(l,j)} \right\}}{\sum\limits_{l=1}^{4} w_{\mathrm{DW}}(l) \cdot \left[ \sum\limits_{j=1}^{9} v_{\mathrm{DW}}(l,j) \right]}, \tag{2}$$

where $w_{\mathrm{DW}}$ are band weights, $\rho_{\mathrm{MISR}}$ are the measured equivalent reflectances from the MISR instrument in band $l$ and camera $j$, $\rho_{\mathrm{m}}$ are the modeled equivalent reflectances from the SMART LUT, $v_{\mathrm{DW}}$ are camera weights (1 if the camera has valid data, 0 if the camera data is potentially contaminated by clouds or sunglint), and $\sigma_{\mathrm{abs,DW}}$ are the total uncertainties. In the
V23 algorithm, the total uncertainties are taken to be 5% of the observed equivalent reflectance for each camera and spectral band (Garay et al., 2020). Finally, the band weights ($w_{\mathrm{DW}}$) are always 1 for the red and near-infrared bands but range from 0 to 1 in the blue and green bands, depending on the AOD being modeled (Martonchik et al., 1998; Kalashnikova et al., 2013).

Although the underlight correction described in Sect. 1 is intended to account for light emitted from deep, clear waters, the spectral band weighting ($w_{\mathrm{DW}}$) is an additional technique used to mitigate significant water-leaving radiance over deep waters,
preferentially seen at shorter wavelengths. If the DW AOD is low (< 0.5), the blue and green bands are excluded from the cost function calculation because water-leaving radiances can be large at shorter wavelengths, even over relatively deep, clear ocean waters (Sayer et al., 2010). However, as the AOD increases the surface contribution to the TOA signal becomes less significant and the blue and green bands are again included.

Now we present the SW approach originally developed by Limbacher and Kahn (2019), which includes modifications to the
DW cost function designed to allow the retrieval to use all four MISR spectral bands in all situations to improve the performance of the retrieval over shallow waters. Details of the derivation of the new cost function can be found in Appendix A. The SW cost function is written as:

$$\chi^2_{\mathrm{abs,SW}}(\tau) = \sum_{l=1}^{4} \sum_{j=1}^{9} \frac{w_{\mathrm{SW}}(j) \cdot \left\{ \rho_{\mathrm{MISR}}(l,j) - \left[ \rho_{\mathrm{m}}(l,j) + \pi \cdot \frac{E_{\mathrm{BOA}}(l)}{E_0(l)} \cdot R_{\mathrm{rs}}(l) \cdot T_{\mathrm{up}}(l,j) \right] \right\}^2}{\sigma^2_{\mathrm{abs,SW}}(l,j) \cdot \left[ \sum\limits_{l=1}^{4} \sum\limits_{j=1}^{9} w_{\mathrm{SW}}(j) \right]}, \tag{3}$$

where $w_{\mathrm{SW}}$ are camera weights, $E_{\mathrm{BOA}}$ are the bottom-of-atmosphere (BOA) downward-directed irradiances, $R_{\mathrm{rs}}$ are the
remote-sensing reflectances, and $T_{\mathrm{up}}$ are the azimuthally-averaged upward transmittances from the surface to the instrument. The additional terms in the numerator's square brackets together are intended to represent the TOA contribution of light reflected by a Lambertian surface with some reflectance ($R_{\mathrm{rs}}$), which is intended to account for any water-leaving radiance contribution from the underwater surface, turbidity, or plant and algae activity. Uncertainties are calculated following Limbacher and Kahn (2019) and include terms for uncertainty in sunglint, the reflectance measurements, and the stray-light correction.
These are then summed in quadrature to get the total uncertainty ($\sigma_{\mathrm{abs,SW}}$). The camera weights ($w_{\mathrm{SW}}$) are initially set to one, but are less than one if there is sunglint present for a given camera (see Limbacher and Kahn, 2019 for more details).



The remote-sensing reflectances, which represent the ratio between the water-leaving radiance and the downwelling surface irradiance (Mobley, 1999), are then calculated for each AOD and mixture by taking the derivative of Eq. 3. This derivative is then set equal to zero and $R_{rs}(l)$ is solved for in order to find the value of $R_{rs}(l)$ at the cost function minimum:

$$
R_{rs}(l) = \frac{\sum_{j=1}^{9} \left\{ \frac{w_{SW}(j)}{\sigma_{abs,SW}^2(l,j)} \cdot T_{up}(l,j) \cdot [\rho_{MISR}(l,j) - \rho_m(l,j)] \right\}}{\pi \cdot \frac{E_{BOA}(l)}{E_0(l)} \cdot \sum_{j=1}^{9} \left[ \frac{w_{SW}(j)}{\sigma_{abs,SW}^2(l,j)} \cdot T_{up}^2(l,j) \right]}.
\tag{4}
$$

A simplified derivation of the analytic solution to the values of $R_{rs}(l)$ can be found in Appendix B. To prevent unphysical negative or zero values of $R_{rs}(l)$, minimum thresholds are used for the four MISR spectral bands: 0.005 (blue), 0.003 (green), 0.0005 (red), and 0.00008 (NIR). Water-leaving radiance is generally higher for shorter wavelengths and lower for longer wavelengths, hence the small minimum threshold values for the red and NIR bands (Sayer et al., 2010).

For the SW retrieval, $\chi_{abs,SW}^2$ is calculated using a unequally-spaced "coarse" 163-point AOD grid. This differs slightly from the operational DW retrieval, which uses a coarse grid of 130 unequally spaced AOD points. In both retrievals, a natural cubic spline is used to interpolate the results to a "fine" grid consisting of 500 AOD points (Witek et al., 2018). In testing the SW algorithm it was determined that an additional 33 grid points were needed at low AODs in the coarse grid to eliminate some numerical artifacts due to the interpolation. The final AOD is then calculated using the "ensemble method" described in Witek et al. (2018). This procedure differs from that of Limbacher and Kahn (2019) in order to make the SW algorithm as consistent as possible with the current operational DW retrieval algorithm, which uses the ensemble method (Garay et al., 2020). As currently implemented, the SW algorithm is designed to only be applied for locations identified as shallow water by the MISR aerosol retrieval.

## 4  Results

### 4.1  AERONET-OC Comparison

The initial comparison between AERONET-OC AOD and the two types of MISR cloud-screened AOD retrievals over shallow waters is shown in Fig. 1. MISR AODs are reported at 550 nm and AERONET-OC AODs were linearly interpolated in log-log space (both axes using logarithmic scales) to the same wavelength using the two nearest available spectral bands (e.g., Witek et al., 2013). The data points are sorted and colored by AERONET-OC measured normalized water-leaving radiance ($L_{WN}$ f/Q) at 667 nm (red) so that points with larger $L_{WN}$ f/Q values lie on top of points with smaller values. This wavelength was selected rather than 551 nm (green), which provided qualitatively similar results, because although the red wavelength has slightly higher uncertainty (Zibordi et al., 2021), it more directly assesses situations where water-leaving radiance may cause issues with the operational MISR DW retrieval that relies primarily on the MISR's 671.1 nm (red) and 866.4 nm (NIR) bands. The comparison was done for both the DW and SW retrieval algorithms in order to assess the performance of the SW retrieval





relative the DW retrieval. Overall, compared to the DW retrievals the SW retrievals have a higher correlation coefficient (R) (0.92 versus 0.87), a smaller root-mean-square error (RMSE) (0.039 versus 0.049), and the SW retrievals are less positively biased (0.0087 versus 0.0126). A larger fraction of the SW retrievals (75.8% versus 68.2%) fall within the expected error envelope (EE), which represents the percentage of retrieved MISR AODs that are within 0.03 or 10% of the corresponding AERONET-OC AOD (e.g., Kahn et al., 2010; Witek et al., 2019). Inspection of Fig. 1a indicates that these statistical differences are driven by a set of points (shown in green/yellow) where the MISR DW AODs are biased high relative to the AERONET-OC retrieved AODs. These same points fall near the dashed one-to-one line for the SW algorithm in Fig. 1b. For the DW algorithm, any water-leaving radiance not accounted for by the surface reflectance and the simple underlight model is treated as being due to the atmospheric path radiance, leading to a higher retrieved AOD. For these high water-leaving radiance cases, the additional Lambertian reflectance term appears to compensate for this effect resulting in better agreement with the AERONET-OC AODs. Note that in cases with low water-leaving radiance, which represent the majority of the dataset, both algorithms perform similarly.

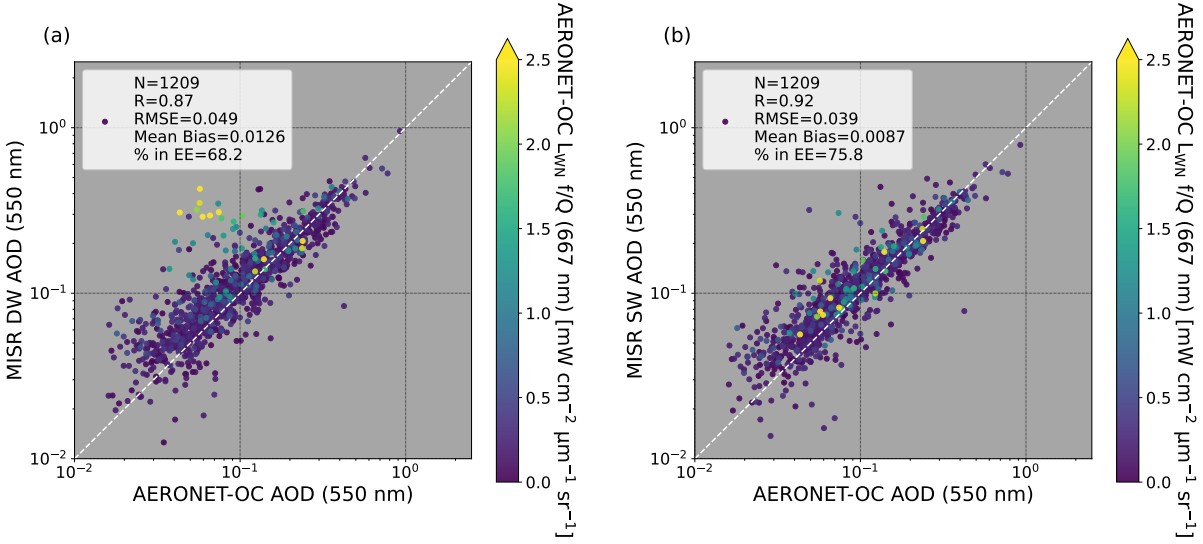

**Figure 1.** Comparison of AERONET-OC AOD to cloud-screened (a) MISR DW AOD and (b) MISR SW AOD. The points are sorted and colored by AERONET-OC normalized water-leaving radiance corrected for bidirectional effects ($L_{WN}$ f/Q) at 667 nm.

Figure 2 shows a comparison of the site-averaged AOD differences between AERONET-OC and MISR DW (left) and AERONET-OC and MISR SW (right). Warmer colors indicate locations where MISR AOD retrievals are generally biased high relative to collocated AERONET-OC AODs, whereas cooler colors show locations where MISR AODs are generally biased low. For the DW retrieval, many of the sites have a high bias, with Lake Okeechobee in Florida and Grizzly Bay in California being particularly notable. Lake Okeechobee has a surface area of 1720 km$^2$, but a mean depth of only 2.7 m, so that frequent resuspension of bottom sediments results in constantly high water turbidity (Canfield et al., 2021). Grizzly Bay





is a subembayment of Suisun Bay near San Francisco, California. The depth of Grizzly Bay is typically less than 3 m and it experiences frequent high turbidity due to wind-wave resuspension of sediment and tidal transport (Bever et al., 2018). When

MISR DW retrievals, which effectively assume the water surface is black, are performed over such shallow waters with high turbidity, the algorithm incorrectly reports an AOD much larger than that measured by AERONET-OC. The MISR SW retrieval, on the other hand, correctly assigns the additional radiance to the surface term and reports AODs in much better agreement with the AERONET-OC measurements. Figure 2 further shows that the agreement between the MISR SW AOD retrievals and the AERONET-OC AOD measurements is improved for nearly every AERONET-OC site, even those not necessarily associated

with extremely shallow waters or constant high turbidity.

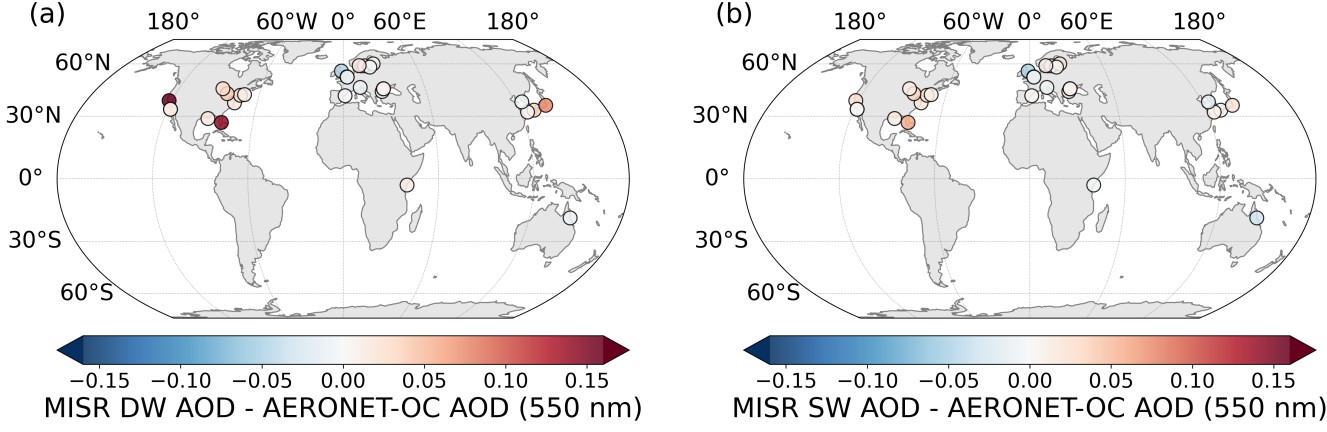

**Figure 2.** Map of site-averaged differences between AERONET-OC AOD and (a) MISR DW AOD and (b) MISR SW AOD.

## 4.2 ARCI Filtering

Figure 1 shows only cloud-screened results over shallow waters. This provides the best overall perspective of the relative performance of the MISR DW and SW AOD retrievals in comparison to AERONET-OC AODs. Operationally, the MISR DW retrieval algorithm provides additional screening based on the Aerosol Retrieval Confidence Index (ARCI) (Witek et al., 2018;

Garay et al., 2020). A low ARCI implies that cost function values are typically large for a given retrieval whereas a high ARCI implies that, for at least some of the 74 mixtures, sufficiently small cost function values are obtained, meaning that the modeled equivalent reflectances are in good agreement with the observed equivalent reflectances. Only those DW retrievals with an ARCI > 0.15 are determined to be good quality. In this section we derive a new ARCI threshold to apply to the SW retrievals over shallow waters. This is done because the DW and SW cost functions differ and thus the ARCI thresholds

are expected to differ as well. Following Witek et al. (2018), Fig. 3 plots the ARCI against the SW cost function ($\chi^2_{\mathrm{abs,SW}}$; Eq. 3) for *all* cloud-screened SW retrievals over shallow waters in the initial 1579 MISR orbits from 23 April 2002 through 30 November 2020 identified as containing an AERONET-OC collocation, not just those retrievals collocated in time and space





with AERONET-OC measurements (as in Sect. 4.1). The horizontal line in both panels shows the manually determined ARCI threshold that separates likely good-quality retrievals from those with lower quality. In particular, note how the ARCI separates

retrievals with unreasonably high AODs (shown in greens/yellow in the right-hand panel) that are correspondingly infrequent (with extremely low counts in the left-hand panel).

As discussed in Witek et al. (2018), this simple screening method with a fixed threshold is highly effective at removing potentially cloud-contaminated retrievals, but it also inadvertently decreases coverage and eliminates some low-AOD retrievals that may agree well with AERONET-OC AODs. However, a more sophisticated screening method would be required to keep

these good quality low-AOD retrievals while still removing high-AOD outliers. The manually selected ARCI threshold for the MISR SW retrieval algorithm was determined to be ARCI > 0.35. This new ARCI threshold removes about 34% of the MISR SW AOD retrievals for this dataset, which is close to the 35.9% of retrievals removed by the ARCI > 0.15 DW threshold used operationally (Witek et al., 2018; Garay et al., 2020).

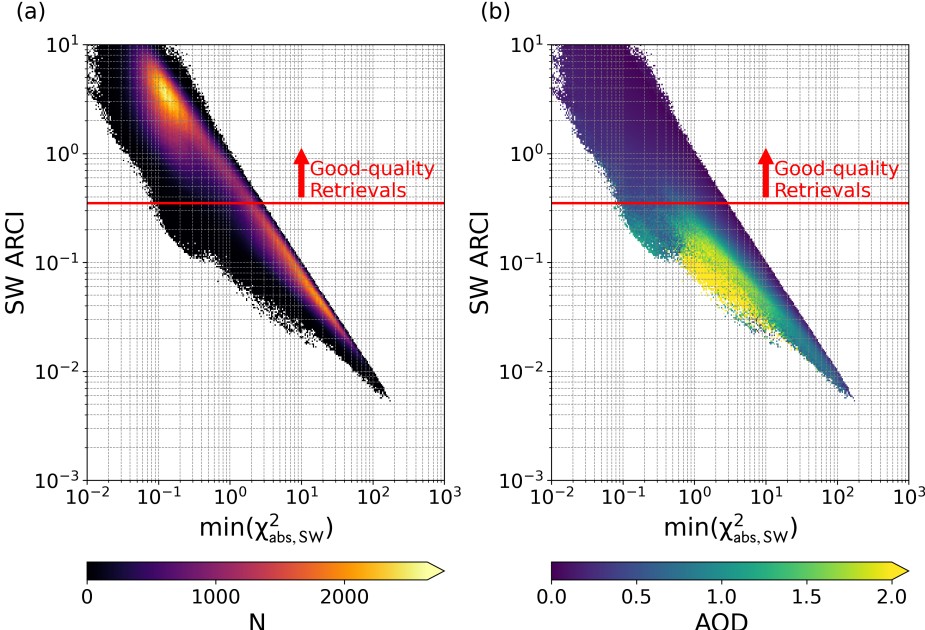

**Figure 3.** (a) Heat map of the minimum SW cost function value versus the SW ARCI, colored by the number of retrievals (N). (b) Heat map of the minimum SW cost function value versus the SW ARCI, colored by retrieved AOD. The red horizontal lines show the new SW ARCI threshold of 0.35 that separates good-quality retrievals from those with poorer quality.

Figure 4 shows a final comparison between MISR AODs and AERONET-OC AODs, after both the cloud screening and ARCI

filtering have been applied. As expected, the ARCI filtering removes many of the high-biased points from the DW retrieval that correspond to high water-leaving radiance in Fig. 1a. This is because the water-leaving radiance results in generally poor agreement with the MISR surface/atmosphere model in the SMART, which effectively assumes a black surface, resulting in





large values in the DW cost function that are filtered out by the ARCI screening. For DW, all the statistics improve at the expense of coverage, which is reduced by 21% (958 matched retrievals compared to the original 1209). The statistics for

the SW retrieval also improve slightly with the ARCI screening, but the reduction in coverage is only 2% (1185 matches compared to 1209). Overall, these results indicate that the SW retrieval algorithm can produce high-quality AODs that agree with AERONET-OC validation data over all types of shallow water scenes, including those with significant water-leaving radiance.

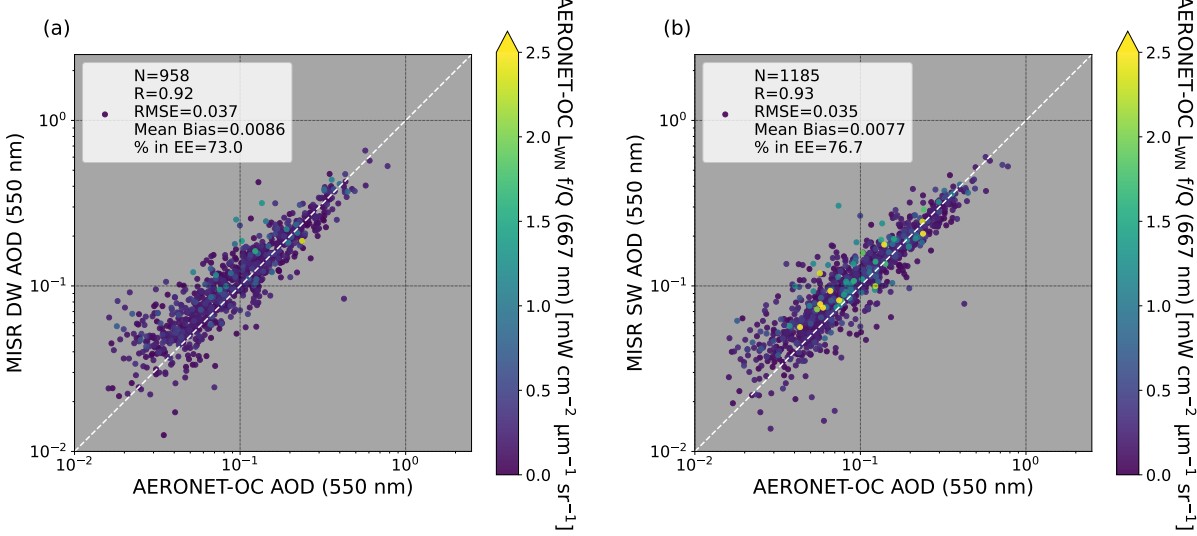

**Figure 4.** Comparison of AERONET-OC AOD to cloud-screened (a) MISR DW AOD with an additional ARCI > 0.15 filter and (b) MISR SW AOD with an additional ARCI > 0.35 filter. The points are sorted and colored by AERONET-OC normalized water-leaving radiance corrected for bidirectional effects ($L_{WN}$ $f/Q$) at 667 nm.

## 4.3 Case Studies

To provide additional context beyond the AERONET-OC comparisons shown above and to check for spatial coherence in the SW retrieval results, three case studies that originally appeared in Limbacher and Kahn (2019) were chosen for further investigation. For each case two setups were used. The first simply reported DW AODs over all water surfaces, regardless of depth or turbidity, whereas the second reported DW AODs over deep waters and SW AODs over shallow waters. In both cases, the operational (V23) land algorithm was used for locations identified as land (Garay et al., 2020). This setup was designed to

assess whether the MISR SW algorithm represents an improvement over DW for shallow water scenes.

The first case is for the Florida Strait on 22 December 2012, shown in Fig. 5. The first panel (Fig. 5a) shows the RGB imagery from the Moderate Resolution Imaging Spectroradiometer (MODIS) instrument, which flies on Terra along with MISR. The wider swath MODIS instrument provides spatial context for the MISR retrievals that lie along the center of the MODIS image.



The second panel (Fig. 5b) indicates the MISR retrieval type, which is based on the assignments in the MISR AGP (Logan,
1999). Locations outside the MISR image swath or that have been cloud screened are labeled as "other", shallow water is
shown in white, whereas land and deep water are depicted as orange and blue, respectively. The third panel (Fig. 5c) shows
the scene with AOD retrievals from the land algorithm over land and the DW algorithm over all locations identified as water.
Finally, the fourth panel (Fig. 5d) shows the AOD retrievals where the MISR SW algorithm is applied over locations designated
as shallow water by the MISR retrieval type. All the AOD retrievals shown only have cloud screening applied so as to provide
more complete coverage of the scene. Operationally, ARCI filtering would also be performed to identify only high-quality
retrievals, but this results in less continuous coverage.

Lake Okeechobee, Florida, described above, with a mean depth of 2.7 m (Canfield et al., 2021), is the primary shallow water
designation in Fig. 5b on the Florida peninsula. In the MODIS image the lake is visibly browner than the deep blue ocean
waters in the center of the image. Water containing significant turbidity (brown) and chlorophyll (lighter blue) are seen north
of the Florida Keys (left central part of the image) and around the Bahamas (right side of the image). Figure 5c shows that
the DW retrievals apparently overestimate the AODs over the shallow waters of Lake Okeechobee and north of the Florida
Keys. On the other hand, the SW retrieval algorithm (Fig. 5d) reports AODs over Lake Okeechobee consistent with the nearby
MISR land retrievals. The SW retrieval algorithm also reduces the AODs north of the Florida Keys, bringing them into better
agreement with the surrounding AODs from the deeper water. However, a small gradient in AOD is seen along the far western
edge of the swath around 25°N, where the water is deeper than 50 m and more than 5 km from land. In Fig. 5b this location
is designated as deep water. Here the AODs from the DW algorithm are reported instead of those from the SW algorithm. Our
testing suggests that these AOD gradients, if present, are generally small and have a minimal impact on the overall accuracy of
the MISR aerosol retrievals. This scene also contains scattered clouds and possibly some cloud-contaminated retrievals on the
eastern edge of the swath. ARCI filtering (not shown) is able to remove some but not all of these erroneously high AODs (see
Sect. 4.2).

The second case study is for the Bohai Sea from 4 January 2015. The panels in Fig. 6 are the same as those in Fig. 5. The
Bohai Sea, along the east coast of mainland China, has a total surface area of 77,000 km$^2$ and an average depth of 18 m. Over
30 rivers drain into the Bohai Sea, including the Yellow River. Because the Yellow River empties into the southwestern part of
the Bohai Sea, the southern portion is typically more turbid than other parts (Xu et al., 2018). This is borne out by the MODIS
270 RGB imagery (Fig. 6a), which shows that on this day the aerosol loading was high and the southern part of the Bohai Sea was
visibly turbid. Given its shallowness, the entire water body is classified as shallow water by the MISR AGP (shown as white in
Fig. 6b), and thus V23 MISR DW aerosol retrievals are not performed operationally. As seen in Fig. 6c, the AODs reported by
the DW algorithm are significantly higher than the MISR AODs retrieved over the nearby land, indicating that the DW AODs
are likely biased high. Some DW retrievals can also been seen along the Yellow River as a thin string of black pixels to the
275 southwest of the Sea, but these are likewise biased high relative to the adjacent land regions, with AODs greater than 1.0. The
SW AODs in Fig. 6d show much better agreement with the nearby MISR land retrievals and the AODs reported for the Yellow
River are reduced to around 0.4. The DW AODs also tend to increase close to the shore and appear to be especially high-biased
over the turbid outflow regions of the Yellow River and the Liao River in the northeast. The SW retrieval algorithm, in contrast,





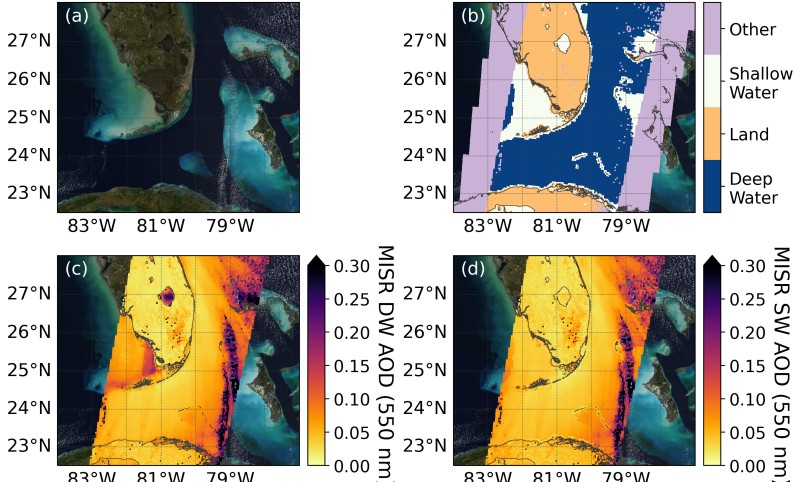

**Figure 5.** The Florida Strait on 22 December 2012, MISR orbit 69220. (a) MODIS-Terra RGB imagery, (b) MISR retrieval type, (c) MISR DW AOD, (d) MISR SW AOD over shallow waters (white in (b)) and MISR DW over deep waters (blue in (b)). No ARCI filtering has been applied. Coastlines are shown in gray.

produces a relatively smooth AOD field over most of the sea that is in much better agreement with the MISR AOD retrievals
280    over land.

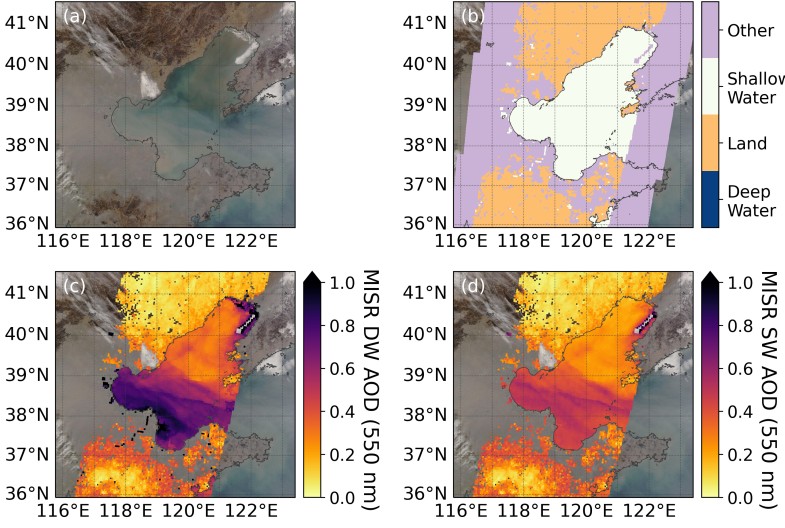

**Figure 6.** The Bohai Sea on 4 January 2015, MISR orbit 80032. (a) MODIS-Terra RGB imagery, (b) MISR retrieval type, (c) MISR DW AOD, (d) MISR SW AOD. No ARCI filtering has been applied. Coastlines are shown in gray.



The third and final case study is for the Caspian Sea on 19 September 2015. The panels in Fig. 7 are similar to those in Figs. 5 and 6, but panel (a) has enhanced brightness and contrast to highlight details. The MISR AGP classifies the entire portion of the Caspian Sea visible in this plot as shallow water, so no V23 operational DW aerosol retrievals are available for the region. However, the bathymetry of the northern and middle Caspian Sea is complicated. The northern portion of the basin is very shallow with an average depth of approximately 5 m (Gunduz and Özsoy, 2014). The Volga River flows into the Caspian Sea from the northwest and the Ural River flows in from the north. Both rivers can be seen in the MODIS RGB image. The transition to the middle Caspian Sea can be seen clearly in Fig. 7a where the green waters change to more blue. The middle Caspian Sea is much deeper, with a mean depth of about 190 m and a maximum depth of 788 m along the western side (Gunduz and Özsoy, 2014). The shallow Garabogazköl Basin, with a depth of less than 10 m can be seen in the extreme bottom right of the MODIS image, but this portion of the Caspian Sea was not imaged by MISR on this date.

An important feature of this case are the plumes from an oil storage fire that extend over the middle Caspian Sea. The source plume itself is visible in the lower right center of the enhanced MODIS image around 43.2°N and 52.4°E. The fire began on 18 September 2015 in the town of Zhanaozen, Kazakhstan. In the MODIS image, dark smoke from this plume is clearly visible over the brighter water. Figure 7c shows that the DW AODs look reasonable, although the pattern of increasing AODs close to the coastline and over river outflow regions in the northern Caspian Sea is evident. However, the DW algorithm captures the enhanced AODs associated with the smoke plume over the water. Again in this case, the SW retrievals (Fig. 7d) mitigate the biases associated with shallow water and turbidity, but the algorithm fails to perform well in the vicinity of the smoke plume. The erroneously low SW AODs in the smoke plume are mostly removed by the ARCI filter (not shown), but this case highlights potential limitations of the SW retrieval algorithm as implemented here.

These three case studies demonstrate that over shallow and turbid waters both the SW and the DW retrieval algorithm perform reasonably well. However, the DW retrieval algorithm often visibly overestimates AODs along coastlines and in river outflow regions, reflecting the influence of increased levels of water-leaving radiance in such areas, which the algorithm interprets as coming from the atmosphere. The SW retrieval algorithm, in contrast, does not exhibit such biases, giving visually smooth AOD retrievals over the majority of shallow waters.

## 4.4 Remote-Sensing Reflectance

Besides AODs, the remote-sensing reflectances ($R_{rs}$) for the Caspian Sea case from Fig. 7 were also compared to an independent remote-sensing reflectance product from Copernicus (Copernicus Climate Change Service, 2022). The Copernicus product combines several different operational satellite products (e.g., MODIS-Aqua, Suomi National Polar-orbiting Partnership Visible Infrared Imaging Radiometer Suite, Sentinel-3A Ocean and Land Colour Instrument) to produce daily composites of $R_{rs}$ on a $0.042°$ by $0.042°$ grid. These remote-sensing reflectances are provided at 412, 443, 490, 510, 560, and 665 nm. Figure 8 compares the MISR $R_{rs}$ from the SW retrieval algorithm to the 560 nm Copernicus $R_{rs}$ product. MISR has a correlation of 0.99 with the Copernicus product and is approximately unbiased, albeit with more scatter at higher $R_{rs}$. Additionally, Fig. 8b demonstrates the enhanced coverage that MISR can provide over regions with significant water-leaving radiance (yellow pixels in the northeastern Caspian Sea).




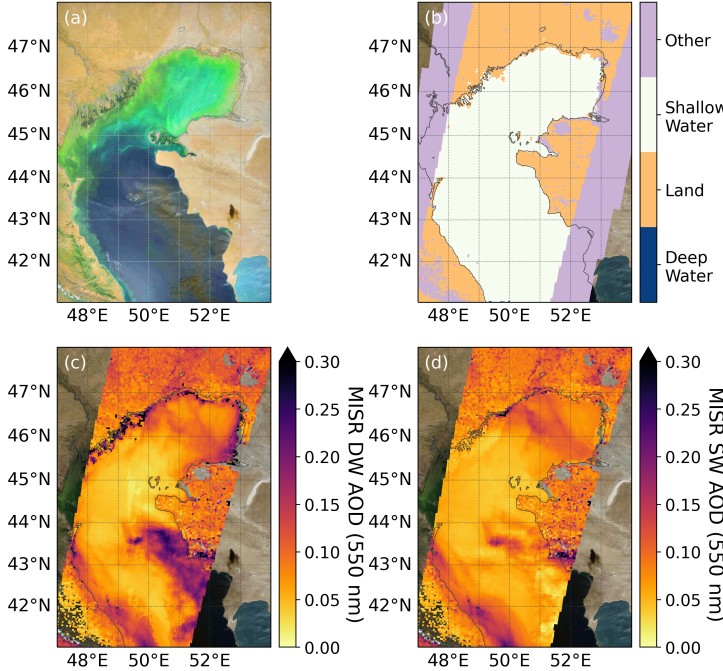

**Figure 7.** The Caspian Sea on 19 September 2015, MISR orbit 83792. (a) MODIS-Terra RGB imagery, brightness and contrast enhanced, (b) MISR retrieval type, (c) MISR DW AOD, (d) MISR SW AOD. No ARCI filtering has been applied. Coastlines are shown in gray.

## 5 Discussion and Conclusions

This work has shown the benefits of a MISR aerosol retrieval algorithm designed for shallow waters, which are scientifically of interest but not currently reported in the operational (V23) retrieval product (Garay et al., 2020). In places with significant water-leaving radiance, MISR DW AODs are often high biased against AERONET-OC AODs because the retrieval assumes that any unaccounted-for brightness in the scene must be from the atmosphere. The SW retrieval algorithm, based on the work of Limbacher and Kahn (2019), includes a Lambertian surface albedo term in each spectral band and is able to account for sunlight reflecting off the underwater surface, turbid waters containing suspended particles, and eutrophic waters that are rich in plant and algae growth. Comparisons to AERONET-OC AODs suggest that using the SW algorithm instead of the DW algorithm for shallow water cases would result in a greater number of high-quality measurements with better statistical agreement with the AERONET-OC dataset. Currently, the SW algorithm has only been implemented for locations identified as shallow water by the MISR aerosol retrieval algorithm. Producing SW retrievals over all water locations, regardless of their designation, would require a greatly expanded validation analysis covering the global oceans where the DW product has been thoroughly validated (Witek et al., 2019), which is beyond the scope of this work.



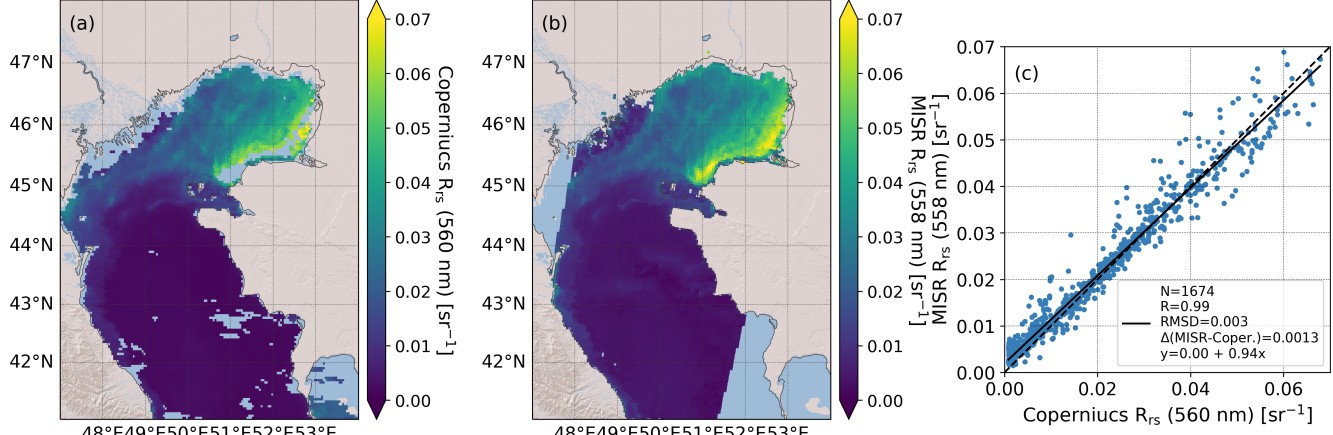

**Figure 8.** The Caspian Sea on 19 September 2015, MISR orbit 83792. (a) Copernicus $R_{\text{rs}}$, (b) MISR $R_{\text{rs}}$, (c) comparison of Copernicus and MISR $R_{\text{rs}}$, created by interpolating both datasets onto a uniform 10 km by 10 km grid. The one-to-one line is shown by the dashed black line and the least squares fit is shown by the solid black line.

One of the limitations of this study is the lack of high AODs in the AERONET-OC comparison. Due to the relative sparsity of the MISR-AERONET-OC collocation dataset over shallow waters, only a handful of cases had AODs of 0.5 or greater.

Because the DW algorithm only uses two spectral bands in low-AOD cases whereas SW always uses all four spectral bands (Sect. 3), this limits the ability to compare the DW and SW retrieval algorithms in cases where the same number of bands are used. The range of AODs could be expanded if the broader AERONET dataset, without water-leaving radiance observations, were to be included in the comparison. For example, Limbacher and Kahn (2019) found 2419 MISR/AERONET collocations on or near coasts over the four years of data they analyzed, although none of the AODs were above 1.0.

Globally, expanding the MISR aerosol product over shallow waters would result in approximately 7% more reported over-water aerosol retrievals than reporting over deep waters only (e.g., 3.16M to 3.40M for June through August 2016). Figure 9 shows that the expanded dataset would result in tens of thousands of additional retrievals each month over biologically productive areas such as coastlines, shallow seas, inland lakes, and large rivers. Examples of regions where data throughput would be significantly increased include the Caspian Sea, the Persian Gulf, and the shallow waters north of Australia.

Future work may include a more extensive validation of the SW retrieval algorithm remote-sensing reflectance beyond the Caspian Sea case shown in this study (Sect. 4.4). This may be valuable to the ocean color community because of MISR's increased information content due to its multi-angle observations and its ability to retrieve $R_{\text{rs}}$ in highly turbid conditions. Additionally, the AOD validation analysis could be expanded to include other sources to increase spatial and temporal coverage (e.g. the Maritime Aerosol Network) and hopefully sample a wider range of AODs. Other MISR aerosol outputs from the SW

retrieval algorithm could also be validated, including Ångström exponents and particle property information.



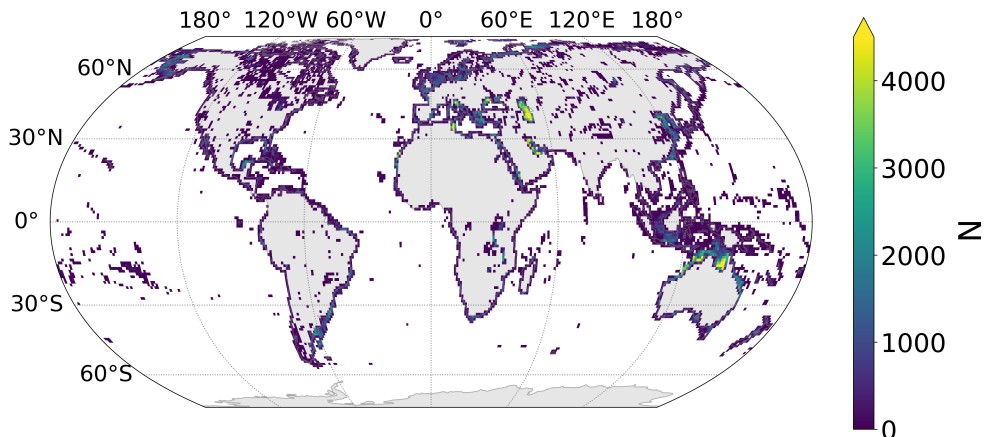

**Figure 9.** Global distribution of the number of MISR retrievals added by including shallow waters (less than 50 m in depth or within 5 km of land) for June through August 2016.

## Appendix A: Radiative Transfer Theory

MISR aerosol retrievals over water are performed using 4.4 km by 4.4 km regions, comprising of 4 by 4 arrays of 1.1 km by 1.1 km subregions. Out of the 16 potential subregions, the single cloud-free pixel with the lowest observed equivalent reflectance ($\rho$) is selected. Observations over this subregion are used as input to the Shallow Water retrieval algorithm.

The SW algorithm uses a physically-based formulation derived from radiative transfer theory for a plane-parallel atmosphere. The governing equation, suitable over shallow, turbid, or eutrophic water surfaces, is derived here, starting from a general case and gradually introducing physically-justified approximations. The TOA reflectances over an arbitrary subregion for a given aerosol mixture and total atmospheric optical depth $\tau_\lambda$ at wavelength $\lambda$ are given by (Diner et al., 2005):

$$
\begin{aligned}
\rho_\lambda^{\mathrm{TOA}}(-\mu, \mu_0, \phi - \phi_0) = {}& \rho_\lambda^{\mathrm{atm}}(-\mu, \mu_0, \phi - \phi_0) \\
& + \left[ \exp(-\tau_\lambda/\mu) r_\lambda(-\mu, \mu_0, \phi - \phi_0) + \int_0^1 \int_0^{2\pi} T_\lambda(-\mu, -\mu', \phi - \phi') r_\lambda(-\mu', \mu_0, \phi' - \phi_0) \mathrm{d}\mu' \mathrm{d}\phi' \right] \\
& \times \frac{1}{E_0} \int_0^1 \int_0^{2\pi} \mu' L_\lambda^{\mathrm{inc}}(\mu', \mu_0, \phi' - \phi_0) \mathrm{d}\mu' \mathrm{d}\phi'
\end{aligned}
\tag{A1}
$$

where $\rho_\lambda^{\mathrm{atm}}$ is the atmospheric path radiance (i.e., the radiance scattered by the atmosphere to space without interacting with the surface), $T_\lambda$ is the diffuse atmospheric bidirectional transmittance function, $L_\lambda^{\mathrm{inc}}$ is the downwelling radiation field incident upon the surface, and $r_\lambda$ is the surface hemispherical-directional reflectance factor (HDRF). Furthermore, $\mu_0$ and $\mu$ are the cosines of the sun and view zenith angles (a negative sign indicates upward direction), and $\phi - \phi_0$ is the view-illumination azimuth angle difference.



In Eq. A1, the hemispheric integral of $L_\lambda^{\mathrm{inc}}$, or the total radiant flux density at the surface, is:

$$E_\lambda(\mu_0) = \int\limits_0^1 \int\limits_0^{2\pi} \mu' L_\lambda^{\mathrm{inc}}(\mu', \mu_0, \phi' - \phi_0) \mathrm{d}\mu' \mathrm{d}\phi' \qquad (A2)$$

This term includes the contribution from all of the multiple reflections between the atmosphere and surface. It is convenient to describe $E_\lambda$ in terms of the idealized radiant flux incident upon a perfectly black body, also called the black surface radiance $E_{\mathrm{BOA}}$, and multiple reflections between the surface and atmosphere. This is done via the highly accurate approximation (exact

for a Lambertian surface) (Liou, 2002; Martonchik et al., 1998):

$$E_\lambda(\mu_0) = \frac{E_{\mathrm{BOA}}(\mu_0)}{1 - A_\lambda(\mu_0)S_\lambda} \qquad (A3)$$

where $S_\lambda$ is the BOA bihemispherical albedo (i.e., the albedo of the atmosphere only), and $A_\lambda$ is the bihemispherical reflectance (BHR), or surface albedo, for non-isotropic incident radiance. Note that in Eq. A3, $E_{\mathrm{BOA}}$ and $S_\lambda$ depend only on atmospheric properties and are pre-calculated in SMART.

Another useful simplification that can be invoked in Eq. A1 is:

$$\int\limits_0^1 \int\limits_0^{2\pi} T_\lambda(-\mu, -\mu', \phi - \phi') r_\lambda(-\mu', \mu_0, \phi' - \phi_0) \mathrm{d}\mu' \mathrm{d}\phi' \approx t_\lambda(-\mu) r_\lambda(-\mu, \mu_0, \phi - \phi_0) \qquad (A4)$$

in which $t_\lambda$ is the upwelling diffuse transmittance of the atmosphere for isotropic radiation, which again depends only on atmospheric properties and is pre-calculated in SMART. Substituting Eq. A3 and Eq. A4 into Eq. A1, we obtain:

$$\rho_\lambda^{\mathrm{TOA}}(-\mu, \mu_0, \phi - \phi_0) = \rho_\lambda^{\mathrm{atm}}(-\mu, \mu_0, \phi - \phi_0) + \frac{1}{E_0} \frac{E_{\mathrm{BOA}}(\mu_0)}{[1 - A_\lambda(\mu_0)S_\lambda]} \left[ exp(-\tau_\lambda/\mu) + t_\lambda(\mu) \right] r_\lambda(-\mu, \mu_0, \phi - \phi_0) \qquad (A5)$$

which shows how signal at TOA over an arbitrary surface depends on atmospheric path radiance ($\rho_\lambda^{\mathrm{atm}}$), atmospheric transmission ($t_\lambda$), and surface HDRF ($r_\lambda$), with the approximations described above. Additional simplifications can be invoked to Eq. A5 for retrievals over shallow waters. First, noting that shallow waters can be assumed to be Lambertian, the $exp(-\tau_\lambda/\mu)$ term is dropped out. Furthermore, assuming that in most conditions both $A_\lambda$ and $S_\lambda$ are relatively small, $(1 - A_\lambda(\mu_0)S_\lambda) \approx 1.0$. These additional approximations lead to:

$$\rho_\lambda^{\mathrm{TOA}}(-\mu, \mu_0, \phi - \phi_0) = \rho_\lambda^{\mathrm{atm}}(-\mu, \mu_0, \phi - \phi_0) + \frac{E_{\mathrm{BOA}}(\mu_0)}{E_0} t_\lambda(\mu) \pi R_{\mathrm{rs}}(-\mu, \mu_0, \phi - \phi_0) \qquad (A6)$$

where the remote-sensing reflectance, $R_{\mathrm{rs}} = r/\pi$, replaces the surface HDRF.





**Appendix B: Derivation of the Analytic Solution to the Values for the Remote-Sensing Reflectance**

Here is a derivation of the analytic solution to the values of the remote-sensing reflectance, $R_{\mathrm{rs}}$. For a set of three cameras, the cost function, $M$ can be written:


$$M = \frac{1}{U_1}\left[\rho_{MI,1} - (\rho_{MO,1} + ERT_1)\right]^2 + \frac{1}{U_2}\left[\rho_{MI,2} - (\rho_{MO,2} + ERT_2)\right]^2 + \frac{1}{U_3}\left[\rho_{MI,3} - (\rho_{MO,3} + ERT_3)\right]^2, \tag{B1}$$

where $U$ represents the uncertainty. Expanding out terms in the first bracket, we get:

$$M = \frac{1}{U_1}\left[\rho_{MI,1}^2 - 2\cdot\rho_{MI,1}\left(\rho_{MO,1} + ERT_1\right) + \left(\rho_{MO,1} + ERT_1\right)^2\right] + \dots \tag{B2}$$

Multiplying out the terms, we get:

$$M = \frac{1}{U_1}\left(\rho_{MI,1}^2 - 2\rho_{MI,1}\cdot\rho_{MO,1} - 2\rho_{MI,1}\cdot ERT_1 + \rho_{MO,1}^2 + 2\rho_{MO,1}\cdot ERT_1 + E^2R^2T_1^2\right) + \dots \tag{B3}$$

Next, we take the partial derivative of $M$ with respect to $R$:

$$\frac{\partial M}{\partial R} = \frac{1}{U_1}\left(0 - 0 - 2\rho_{MI,1}\cdot ET_1 + 0 + 2\rho_{MO,1}\cdot ET_1 + 2E^2T_1^2R\right) + \dots \tag{B4}$$

Setting this to zero, simplifying slightly, and including all the cameras we get:

$$0 = \frac{2ET_1}{U_1}\left(-\rho_{MI,1} + \rho_{MO,1} + ET_1R\right) + \frac{2ET_2}{U_2}\left(-\rho_{MI,2} + \rho_{MO,2} + ET_2R\right) + \frac{2ET_3}{U_3}\left(-\rho_{MI,3} + \rho_{MO,3} + ET_3R\right) \tag{B5}$$

The factor of $2E$ cancels, and we can rearrange the equation to get:


$$\frac{T_1}{U_1}\left(\rho_{MI,1} - \rho_{MO,1}\right) + \frac{T_2}{U_2}\left(\rho_{MI,2} - \rho_{MO,2}\right) + \frac{T_3}{U_3}\left(\rho_{MI,3} - \rho_{MO,3}\right) = RE\left(\frac{T_1^2}{U_1} + \frac{T_2^2}{U_2} + \frac{T_3^2}{U_3}\right) \tag{B6}$$

Now, solving for $R$ and recognizing places we can use summation, we get:

$$R = \frac{\sum_i \frac{T_i}{U_i}\left(\rho_{MI,i} - \rho_{MO,i}\right)}{E\sum_i \frac{T_i^2}{U_i}}, \tag{B7}$$

which is just Eq. (5) for $R_{\mathrm{rs}}$.

*Code and data availability.* MISR data can be downloaded in NetCDF format from the NASA Langley Research Center Atmospheric Sci-
ence Data Center: https://asdc.larc.nasa.gov/project/MISR (last access: 8 March 2023). AERONET-OC data used in this study was download



in June 2021 from the AERONET website available at: https://aeronet.gsfc.nasa.gov/new_web/download_all_v3_lwn.html (last access: 8 March 2023).

*Author contributions.* RN led the project. MW and MG helped supervise the project and MW prepared the AERONET-OC comparison dataset. MB performed the retrievals. JA, RK, and DD advised on the methodology and helped interpret the results. RN wrote the manuscript
with contributions from all authors.

*Competing interests.* The authors declare that they have no conflict of interest.

*Acknowledgements.* The research was carried out at the Jet Propulsion Laboratory, California Institute of Technology, under a contract with the National Aeronautics and Space Administration (80NM0018D0004). Support from the MISR project is acknowledged. We thank the AERONET-OC PIs and Co-Is and their staff for establishing and maintaining the 31 sites used in this investigation.



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
