# Peer review of "Expanding the coverage of MISR aerosol retrievals over shallow, turbid, and eutrophic waters"

_EGUsphere, 2023_

## Author Comment (AC1)

The authors would like to thank Anonymous Referee #1 for their comments on our manuscript entitled, "Expanding the coverage of MISR aerosol retrievals over shallow, turbid, and eutrophic waters." Below, we have addressed their comments and made the necessary changes in the manuscript.

**"How BOA downward-directed irradiances and azimuthally-averaged upward transmittances from surface to the instrument are obtained in Equation (4) need to be explained. The two terms are affected by AOD, but AOD is unknown when it is to calculate remote-sensing reflectance in Equation (4)."**

$E_{BOA}$ and $T_{up}$ are pre-computed and provided by the SMART look-up table for the 74 mixtures and 130 (or 163 for SW) AODs of interest. They are then used when calculating $R_{rs}(l)$ in Equation (4). We have added this statement to Section 3 (L141):

"$E_{BOA}$ and $T_{up}$ are, like $\rho_m$, pre-computed and provided by the SMART LUT for each of the 74 mixtures and AODs ranging from 0 to 3."

**"In Fig. 1 and Fig. 2, AERONET-OC normalized water-leaving radiance corrected for bidirectional effects (LWN f/Q) at 667 nm. I suggest changing it to be surface reflectance (or remote-sensing reflectance called in manuscript); this may be better to demonstrate the impact of water-leaving reflectance on retrieving AOD quantitively."**

The remote-sensing reflectance ($R_{rs}$) is equal to the normalized water-leaving radiance corrected for bidirectional effects ($L_{wn}fQ$) divided by the exo-atmospheric solar irradiance at 1 AU ($E_0$):

$$R_{rs}(\lambda) = L_{WN}fQ(\lambda)/E_0(\lambda)$$

See a detailed explanation here, specifically equation 10:
https://www.oceanopticsbook.info/view/atmospheric-correction/normalized-reflectances

Thus, the resulting Figs. 1 and 2 would be very similar to the current figures if AERONET $R_{rs}$ was used instead of $L_{wn}fQ$ because $E_0$ is approximately constant.

**"In Fig. 5d, the transition from AOD retrieved by DW to SW is not very smooth at around 82.5W, 25N; DW AOD over deep water is larger than SW AOD over shallow. At 81.5, 25.7N, there are some hotspots. All these should be discussed."**

The transition from AOD retrieved by DW to SW around 25° N, 82.5° W is discussed on L260:

"However, a small gradient in AOD is seen along the far western edge of the swath around 25° N, where the water is deeper than 50 m and more than 5 km from land. In Fig. 5b this location is

designated as deep water. Here the AODs from the DW algorithm are reported instead of those from the SW algorithm. Our testing suggests that these AOD gradients, if present, are generally small and have a minimal impact on the overall accuracy of the MISR aerosol retrievals."

The AOD hotspot at 25.7° N, 81.5° W in Fig. 5d is likely real. It is in both the DW and SW AOD plots and the MISR DF camera has enhanced radiance in that area, suggesting a real aerosol feature:

[Figure]

https://l0dup05.larc.nasa.gov/MISR_BROWSE/time

Additionally, if you toggle between MODIS Terra and Aqua visible imagery, it appears there may be white smoke just to the NE of the MISR AOD hotspot, around 25.91° N, 81.52° W.

---

## Author Comment (AC2)

The authors would like to thank Anonymous Referee #2 for their comments on our manuscript entitled, "Expanding the coverage of MISR aerosol retrievals over shallow, turbid, and eutrophic waters." Below, we have addressed their comments and made the necessary changes in the manuscript.

**"L103: It sounds odd for an algorithm to 'contain' something; perhaps 'considers'."**

We indented to mean that the LUT, not the algorithm, contains 74 mixtures and have made the following clarification:

"The current operational retrieval algorithm (V23) LUT effectively contains 74 unique aerosol mixtures"

**"Eq.2: There is a \tau on the LHS of this equation but not on the RHS. I believe \rho_m is the function of AOD. (Same the same is true of later equations.)"**

For clarity we have remove the ($\tau$) on the LHS of Eqs. 2 and 3. The text states that they are both calculated at several mixtures and AODs.

**"L416: Berenfeld et al. 2005 is article number GB1006." and others**

It looks like article numbers shouldn't be included in AMT references:

"Please supply the full author list with last name followed by initials. After the list of authors, the complete reference title needs to be named. Journal names are abbreviated according to the Journal Title Abbreviations by Caltech Library, followed by the volume number, the complete page numbers (first and last page), the digital object identifier (DOI), and the publication year."

https://www.atmospheric-measurement-techniques.net/submission.html#references